# How Progesterone Receptor Expression Impacts Platinum Sensitivity in Ovarian Clear Cell Carcinoma: Insights from Clinical and Experimental Perspectives

**DOI:** 10.3390/ijms25147942

**Published:** 2024-07-20

**Authors:** Chen-Hsuan Wu, Hung-Chun Fu, Yu-Che Ou, I-Chieh Chuang, Jui Lan, Ming-Yu Yang, Hao Lin

**Affiliations:** 1Department of Obstetrics and Gynecology, Kaohsiung Chang Gung Memorial Hospital, Chang Gung University College of Medicine, Kaohsiung 833, Taiwan; chenhsuan@cgmh.org.tw (C.-H.W.); allen133@cgmh.org.tw (H.-C.F.); ou4727@cgmh.org.tw (Y.-C.O.); 2Graduate Institute of Clinical Medical Sciences, College of Medicine, Chang Gung University, Taoyuan 333, Taiwan; yangmy@mail.cgu.edu.tw; 3Department of Obstetrics and Gynecology, Chia-Yi Chang Gung Memorial Hospital, Chiayi 613, Taiwan; 4Department of Anatomic Pathology, Kaohsiung Chang Gung Memorial Hospital, Chang Gung University College of Medicine, Kaohsiung 883, Taiwan; b9205043@cgmh.org.tw (I.-C.C.); blueray@cgmh.org.tw (J.L.); 5Department of Otolaryngology, Kaohsiung Chang Gung Memorial Hospital, Chang Gung University College of Medicine, Kaohsiung 883, Taiwan

**Keywords:** ovarian clear cell carcinoma, progesterone receptor, platinum sensitivity

## Abstract

Ovarian clear cell carcinoma (OCCC) is often considered a relatively platinum-resistant malignancy. The aim of this study was to explore the influence of progesterone receptor (PR) expression levels on platinum sensitivity and survival outcomes in people with OCCC. A retrospective analysis was conducted with 80 people with OCCC who underwent surgery followed by adjuvant chemotherapy. PR expression was assessed via immunohistochemical (IHC) staining and quantified using the H score. The platinum sensitivity and survival outcomes of patients with weak and strong PR expression were compared. Additionally, cisplatin viability and migration experiments were conducted with OCCC cell lines (ES-2 and TOV-21G) with varying PR isoform expressions. Among the 80 patients, 62 were classified as having platinum-sensitive disease, while 18 had platinum-resistant disease. The mean total PR H- score of platinum-sensitive tumors was significantly higher than that of platinum-resistant tumors (*p* = 0.002). Although no significant differences in progression-free and overall survival were observed between patients with high and low PR expression, those with high PR expression tended to have longer survival. While PR protein was only weakly detectable in ES-2 and TOV-21G cells, a transfection of the PR-A or PR-B gene resulted in a strong expression of PR-A or PR-B, which led to significantly reduced proliferation and migration in ES-2 and TOV-21G cells. Furthermore, overexpression of PR-A or PR-B enhanced cisplatin cytotoxicity in these cell lines. In conclusion, strong PR expression was associated with improved platinum sensitivity and survival outcomes, consistent with our experimental findings. The potential of PR as a tumor sensitizer to cisplatin in OCCC warrants further investigation.

## 1. Introduction

According to a global study published in 2022, approximately 313,959 people were newly diagnosed with ovarian cancer globally with 207,252 mortalities from this disease in 2020, making this malignancy the most lethal gynecological cancer [1]. Meanwhile, the cancer statistics of the Taiwan Cancer Registry revealed 1793 newly diagnosed cases of ovarian cancer and 696 deaths in 2021, ranking it as the seventh most common cause of cancer death of women [2].

The majority of ovarian malignancies comprise epithelial ovarian cancer (EOC), with various incidence rates of EOC subtypes noted across regions. The incidence of ovarian clear cell carcinoma (OCCC) in Asia ranges from 13% to 28%, which is significantly higher than the Western incidence of 5–10% [3,4]. In Taiwan, OCCC accounted for about 19.4% of all EOC cases during the period between 2002 and 2015 [5]. OCCC stands out due to its unique cellular and molecular characteristics, and endometriosis is recognized as a precancerous lesion with a three-fold increased risk of OCCC development. OCCC is frequently diagnosed at an early stage and exhibits a favorable prognosis. However, advanced disease is considered a platinum-resistant malignancy with a poorer prognosis compared to high-grade serous carcinoma [6,7,8,9]. In more recent studies, it was revealed that cisplatin resistance in EOC primarily results from the enhanced antioxidant capacity of ovarian cancer cells [10]. Therefore, there is an urgent need for novel treatments, prompting extensive research into molecular targeted therapy and immunotherapy.

The presence of sex steroid hormone receptors in many EOC tissues suggests a potential role for hormones in the etiology and progression of this disease [11]. Epidemiological evidence strongly indicates that progesterone-containing contraceptives and pregnancy may exert a protective effect against EOC development, whereas postmenopausal status without progesterone protection increases incidence, highlighting the significance of the progesterone receptor (PR) in prognostication [12]. While studies have shown that strong PR expression correlates with improved disease-specific survival in patients with ovarian high-grade serous and endometrioid cancer, OCCC findings have been inconclusive due to limited sample sizes and statistical power [13,14]. However, the differential expression of PR from atypical endometriosis to OCCC has been reported, suggesting its involvement in tumorigenesis [15,16]. Furthermore, the association between PR status and chemosensitivity remains unexplored. Therefore, this study was conducted to investigate the relationship between PR expression, platinum sensitivity, and survival outcomes in both people with OCCC and cell-line models.

## 2. Results

### 2.1. Clinical Data

During the study period, a total of 592 ovarian cancer patients were examined for eligibility. Patients were excluded for various reasons, including 481 patients for non-OCCC histology, 15 patients for inadequate or insufficient chemotherapy, 9 patients for insufficient tissue for immunohistochemical (IHC) staining, and 7 patients for inadequate follow-up or presence of double cancer. Ultimately, 80 patients met the eligibility criteria and formed this study’s cohort. The mean age of subjects at diagnosis was 48.9 years (a range of 28–61 years), with a median follow-up time of 50 months (a range of 5–132 months). The basic characteristics of the patients are detailed in Table 1. Among the 80 patients recruited, 66 were classified as having platinum-sensitive disease, while 14 were classified as platinum-resistant. Platinum-sensitive patients exhibited a significantly higher rate of concurrent endometriosis and a higher mean PR H score, as indicated in Table 2. Positive PR staining (H score > 1) was observed in 30 patients (37.5%). Utilizing receiver operating characteristic (ROC) curve analysis, the optimal cut-off value of the PR H score for predicting platinum sensitivity was determined to be 50 (AUC 0.64, 95% CI: 0.51–0.77). Strong PR expression was defined as an H score of 50 or higher, while weak PR expression was defined as a score lower than 50. No differences in the distribution of clinicopathological variables were observed between patients with strong and weak PR expression, as summarized in Table 3. Representative examples of IHC staining of PR in OCCC are presented in Figure 1, with a positive control from breast cancer tissue provided in Appendix A. Univariate analysis revealed significantly improved 5-year progression-free survival (PFS) and overall survival (OS) for patients with early International Federation of Gynecology and Obstetrics (FIGO) stage (I/II) and concurrent endometriosis (Appendix A). Multivariate Cox regression analysis indicated that only an advanced FIGO stage was a significant factor associated with worse PFS (HR 2.427; 95% CI 1.004–5.869) and OS (HR 3.22; 95% CI 1.27–8.161) (Table 4). Although no statistically significant differences in survival were found between weak and strong PR expression groups, we observed higher PFS and OS rates in the strong PR group (Appendix A). As the FIGO stage was the sole independent predictor of survival, the correlation between PR expression and FIGO stage was found to show a stepwise decrease, although not statistically significant, in strong PR expression from FIGO stage I to IV (Table 3).

### 2.2. Cell Line Data (Cell Viability Assay)

Immunoblot detection of PR-A and PR-B exhibited almost undetectable signals in both ES-2 and TOV-21G wild-type cells, as illustrated in Figure 2A,B. Following transfection with plasmids pcDNA3-vector, pcDNA3-PR-A or pcDNA3-PR-B, immunoblotting revealed approximately a 5-fold increase in PR-A protein expression in both ES-2-PR-A and TOV-21G-PR-A over-expression cells when compared to ES-2- and TOV-21G-vector cells, while ES-2-PR-B- and TOV-21G-PR-B-overexpressing cells also exhibited a significant elevation (about a 14-fold increase) in PR-B protein levels relative those of ES-2- and TOV-21G-vector cells (Figure 2C,D). Cell viability assays indicated that ES-2 cells overexpressing PR-A and PR-B displayed significantly slower proliferation rates than cells transfected with vector only, while statistical significance was only observed in PR-B-overexpressing TOV-21G cells (Figure 2E,F). Furthermore, the cytotoxic effects of cisplatin on ES-2 and TOV-21G cells with varying PR-A and PR-B expression levels were assessed. The IC50 (50% inhibitory concentration) values of cisplatin in our cell lines were determined to be 3.34 μM for both ES-2 and TOV-21G. As compared to the control (vector) cells, cisplatin-induced inhibition of proliferation was significantly pronounced in both PR-A- and PR-B-overexpressing ES-2 cells (Figure 2G). Similar results were noted in TOV-21G cells, with an exception that only PR-B-overexpressing cells exhibited significantly slower proliferation rate at 96 h compared to vector cells (Figure 2H). We observed a greater improvement in chemosensitivity in the ES-2 cell line compared to TOV-21G (Figure 2G,H), suggesting that the overexpression of PR may enhance drug sensitivity more in chemo-resistant cells such as ES-2.

### 2.3. Cell Line Data (Transwell Migration Assay)

Upon the confirmation of the cytotoxic effect of cisplatin in ovarian clear cell carcinoma (OCCC) cells, its impact on cancer cell migration was then investigated with a Transwell assay. The results revealed that ES-2 or TOV-21G cells overexpressing PR-A or PR-B exhibited significantly reduced migration ability compared to that of control cells. Furthermore, treatment with cisplatin led to a more pronounced inhibition of metastatic capacity in PR-A- or PR-B-overexpressing OCCC cells, as evidenced by a significant decrease in the number of migrated cells per ×200 field (Figure 3A,B).

## 3. Discussion

Our findings revealed a correlation between platinum-sensitive tumors and enhanced PR expression, observed in both clinical specimens and cell line models. However, while PR status did not exhibit significant association with clinical outcomes, there was a notable trend indicating improved PFS and OS among patients with OCCC.

Previous studies examining sex hormone receptor expression rates in patients with OCCC have reported a broad range of PR expression, ranging from 0% to 60% [17,18], with most falling below 10% positivity [19,20,21,22]. In our study, a relatively higher percentage, at 37.5%, of patients tested positive for PR. Discrepancies among studies may stem from the various PR antibodies used for IHC staining. With two major isoforms, PR-A and PR-B, the commonly used commercial PR antibodies include clones 16, 636, 1A6, alpha PR6, and 1E2. Despite claims of recognizing both PR-A and PR-B isoforms, prior research has demonstrated that certain antibodies may not effectively detect PR-B in tissue sections using IHC techniques, even though they show efficacy in immunoblot analysis [23]. This discrepancy may arise from the masking of PR-B epitopes during the heat treatment of formalin-fixed tissue, causing a degree of tertiary structural change in the protein. Consequently, PR expression may be undetected or underestimated, particularly in tissues where PR-B is predominant.

To date, no study has assessed PR expression exclusively in people with OCCC. In the largest collaborative study by Sieh et al., it was revealed that PR expression correlated with improved survival in high-grade serous and endometrioid carcinoma but not in mucinous or OCCC cases [13]. Conversely, a meta-analysis of 5685 people with EOC from 28 studies indicated a relationship between PR expression and either PFS or OS. However, when focusing on serous carcinoma studies, the prognostic significance of PR vanished, contradicting the findings of Sieh et al. [14]. This disparity may stem from variations in PR measurement methodology and interpretation criteria for IHC results. Different from Sieh et al., the H score was utilized for PR quantification in this study. Previous studies commonly employed a three-tier system—negative, weak, and strong—where positivity was defined as staining in <1%, 1–50%, and ≥50% of tumor cell nuclei, respectively. In our study, a sequential decrease in strong PR expression from FIGO stage I to IV was observed. Our data with cell lines also revealed that cells overexpressing PR exhibited significantly reduced migration compared to cells lacking PR expression. These findings align with earlier observations of various levels of PR expression, ranging from 100% in atypical ovarian endometriosis to 35% in OCCC, suggesting PR loss may contribute to OCCC etiology and progression [15,16]. Despite extensive efforts to determine an optimal PR cutoff value, the feasibility of PR in acting as a significant prognostic factor has been disapproved. Several factors may explain this outcome. Firstly, our cohort included many FIGO stage I/II patients, known for their favorable prognosis, potentially obscuring PR’s prognostic utility. Secondly, although PR was quantified, H-score values were not normally distributed. Attempts to stratify patients by stage, concurrent endometriosis status, or transformation of values yielded no conclusion. Finally, the exclusion of a certain number of cases due to various reasons may have introduced bias into the outcomes.

The specific roles of each PR isoform in hormone treatment responsiveness and prognosis remain inconclusive. In breast cancer, PR-A-predominant tumors showed greater sensitivity to antiprogestin treatment, whereas PR-B-predominant tumors were linked to advanced disease and poorer prognosis [24]. Conversely, in endometrial cancer, both PR-A and PR-B were associated with reduced tumor aggressiveness, but only PR-B predicted a better prognosis [25]. Notably, our study revealed that the presence of either PR isoform A or B seemed to enhance sensitivity to cisplatin, with patients exhibiting higher total PR expression, showing a trend toward improved survival. The relationship between hormone receptor status and platinum sensitivity has seldom been explored in EOC. In one preclinical study using OVCAR-3 cell, Peluso et al. demonstrated that high PR expression in ovarian cancer cells correlated with decreased expression of PR membrane component-1 (PRMC-1), thereby enhancing cisplatin effectiveness [26]. The same group later transplanted SKOV-3 cells into nude mice, which resulted in the development of the OCCC animal model. They demonstrated that PRMC-1 regulated not only the growth and platinum sensitivity of cultured SKOV-3 cells but also the ovarian tumors derived from these cells [27]. Additionally, in an endometrial tumor model, they observed that PRMC-1-depleted tumors displayed heightened responsiveness to chemotherapeutic stress compared to PRMC-1-intact control tumors [28]. All these data support the close relationship between PRMC-1 expression and chemosensitivity. Given that there i an inverse relationship between the PRMC-1 and PR expression [26], our results align with Peluso et al.’s findings, although further investigation into the relationship between PRMC-1 and PR with different isoform expressions is warranted.

Ovarian clear cell carcinoma has long been considered a platinum-resistant malignancy. Although there is currently no highly accurate biomarker for predicting chemotherapy sensitivity before treatment, our findings highlight the necessity of developing alternative therapeutic strategies for patients with low PR expression. Recent research has identified specific immune-related molecular profiles, including PD-L1 expression and mismatch repair protein defects, in OCCC [29,30]. These findings have prompted the development of novel therapeutic strategies focusing on immunotherapy. While early phase I/II trial results with immune check point inhibitors showed promise, the sample sizes were limited, necessitating further validation in larger phase II/III trials [31,32]. Until comprehensive clinical data are available, clinicians must continue with great efforts to identify patients who may gain genuine benefits from traditional chemotherapy.

While our study provides valuable insights into the influence of PR expression on platinum sensitivity and survival outcomes in people with OCCC, several limitations should be acknowledged. Firstly, the sample size of 80 people with OCCC may limit the generalizability of our findings, and larger cohorts with a focus on advanced stage disease are needed to validate our results. Additionally, the investigation of only two OCCC cell lines (ES-2 and TOV-21G) may not have fully captured the heterogeneity of OCCC, and future studies should consider exploring a broader range of cell lines. Furthermore, the lack of animal models and mechanistic studies hindered our ability to elucidate the underlying pathways through which PR expression influences platinum sensitivity.

## 4. Materials and Methods

### 4.1. Patients

A retrospective review was conducted with people diagnosed with OCCC between January 2008 and December 2016 at Kaohsiung Chang Gung Memorial Hospital. Patients who underwent primary surgery followed by at least 4 cycles of adjuvant chemotherapy with an availability of tissue for immunohistochemical (IHC) staining were recruited. Staging was determined in accordance with the International Federation of Gynecology and Obstetrics (FIGO) 2014 system. Cases with insufficient tissue for analysis, inadequate chemotherapy (<4 cycles), lack of regular follow-up, or presence of double cancers were excluded. Clinical data, including demographics, FIGO staging, carbohydrate antigen-125 (CA-125) levels, concurrent endometriosis (defined as endometriosis coexisting with the cancer site), platinum sensitivity, PR expression, and survival duration, were extracted from clinical records. Platinum sensitivity was classified as resistant or sensitive based on the time elapsed since completion of the first-line chemotherapy (with a cutoff of 6 months). Clinical parameters, including PR expression, were compared between platinum-sensitive and platinum-resistant patients. This study’s protocols were approved by the Institutional Review Board of Chang Gung Memorial Hospital (IRB approval number: 201601487B0C503).

### 4.2. Immunohistochemistry Analysis

Currently, as the College of American Pathologists (CAP) does not specifically outline protocols for assessing PR IHC staining in ovarian cancer, we adopted a protocol similar to that recommended for breast cancer based on CAP guidelines. Briefly, paraffin-embedded ovarian tissues were sliced into 3 μm-thick sections, which were deparaffinized in xylene and rehydrated through an alcohol gradient. Activity of endogenous peroxide was blocked with 3% H_2_O_2_ for 15 min. Afterward, slides were treated with 10× citrate buffer pH6.0 (Sigma-Aldrich, St. Louis, MO, USA C9999) for 17 min in a microwave oven for antigen retrieval and then incubated with the primary antibody PR (1:100, Leica, Deer Park, TX, USA. Cat No. #PR NCL-L-PGR-312) at 4 °C overnight, which can recognize both A and B isoforms of PR. IHC staining was performed using an UltraVision Quanto Detection System HRP kit (ThermoFisher, Fremont, CA, USA, TL-060-QHL) and DAB Quanto reagent (ThermoFisher, Fremont, CA, USA, TA-125-QHDX) according to the manufacturer’s instructions. Finally, the sections were counterstained with hematoxylin and mounted with aqueous mounting media. Slides were visualized at 200× magnification and PR staining was scored with an H-scoring system (ranging from 0 to 300) by multiplying the tumor nuclei cell intensity (on a scale of 0 to 3) by the percentage of positive tumor nuclei cells (on a scale of 0 to 100). All the cases were blindly scored by two pathologists (Chuang IC, Lan J). Since normal breast epithelial cells have receptors for estrogen and progesterone and proliferate under their influence, we used breast tissue for positive control of PR in our study. Our pathology laboratory is accredited under the CAP Laboratory Accreditation Program (CAP number 7232636), demonstrating compliance with rigorous quality and performance standards established by the CAP.

### 4.3. PR-Overexpressing Ovarian Carcer Cell Lines

We selected ES-2 and TOV-21G as OCCC cell lines for our drug sensitivity experiments due to their distinct characteristics. ES-2 cells are known to be resistant to cisplatin, whereas TOV-21G cells are reported to be susceptible to it [33]. The plasmids pcDNA3-empty vector, pcDNA3-hPR-A (Cat. No. #89119) and pcDNA3-PR-B (Cat. No. #89130) were acquired from Addgene, Watertown, MA, USA. Prior to transfection, ES-2 and TOV-21G cells were seeded in 6-well plates at a density of 3 × 10^5^ cells/well and then incubated for 24 h. Transfection was then performed using Lipofectamine^TM^ 2000 reagent (Invitrogen^TM^, Thermo Fisher Scientific Inc., Fremont, MA, USA, Cat. No. #11668019) following the manufacturer’s instructions, together with negative-control cells to eliminate any potential effects of transfection on cell viability. After a 48 h incubation period and a confirmation of successful transfection via Western blotting, samples were subjected to analyses for cell viability and migration.

### 4.4. Western Blot Analysis

Cell proteins were lysed in PRO-PREP Protein Extraction Solution (iNtRON, Cat. No. #17081.1), and the protein concentrations in the lysates were determined using a BCA Protein Assay kit (Merck, Millipore, Burlington, MA, USA, 71285-3). Subsequently, the proteins were separated by 8–10% SDS-PAGE and transferred onto PVDF membranes (Merck Millipore, IPVH00010). The membranes were then blocked with 5% nonfat milk in Tris Buffered Saline (SIGMA, T5912) containing 0.1% Tween-20 for 30 min at room temperature. Next, the membranes were probed with specific primary antibodies: progesterone receptor isoform A/B (D8Q2J) XP^R^ rabbit mAb (1:1000, Cell Signaling, Danvers, MA, USA, Cat. No. #8757), and β-actin (1:10,000, Merck Millipore, Cat. No. #A5441) at 4 °C overnight. Following incubation with the primary antibodies, the membranes were incubated with a secondary antibody conjugated with horseradish peroxidase (HRP) (1:8000, Merck Millipore, Cat. No. #A9044) for 60 min at room temperature. Immunoreactive bands were visualized using Immobilon Western Chemiluminescent HRP Substrate (Merck Millipore, Cat. No. #WBKLS0500) and exposed to X-ray film for autoradiography. Densitometry was performed using ImageJ V1.5.3K.

### 4.5. Cell Viability Assay

We seeded 8 × 10^3^ cells (TOV-21G) per well in 24-well cell-culture plates with the indicated growth medium and seeded 1 × 10^4^ cells (ES-2) per well in 24-well cell-culture plates with the indicated growth medium. After 24 h, cisplatin was added, and then the cells were trypsinized after 24, 48, 72 and 96 h, and trypan blue dye was used to determine the number of viable cells present in a cell suspension, upon which the cell number was counted to produce a cell-growth curve.

### 4.6. Cell Transwell Migration Assay

The ovarian cancer cell suspensions were prepared in serum-free medium and seeded onto a porous membrane in the upper chamber of a Transwell insert, allowing them to migrate toward a chemoattractant present in the lower chamber. We utilized SPL Life Sciences (Pocheon-si, Republic of Korea) Transwell^®^ (specifically, the 6.5 mm Transwell^®^ with 8.0 µm Pore Polycarbonate Membrane Insert), identified by code number 35224. The Transwell units were filled with growth medium and incubated at 37 °C for 24 h. At the end of incubation, nonmigrated cells were removed, and the migrated cells on the underside of the membrane were fixed and stained with 0.1% crystal violet at room temperature before enumeration under a microscope. Data are expressed as mean ± SD.

### 4.7. Statistical Analysis

Receiver operating characteristic (ROC) curve analysis was conducted to determine the optimal cut-off value of the PR H score for predicting platinum sensitivity. Progression-free survival (PFS) and overall survival (OS) were defined as the duration from the date of diagnosis to the first evidence of progression and death, respectively. Median and mean values were compared using the independent two-sample *t*-test, while frequency distributions among categorical variables were compared using the chi-square or Fisher’s exact test. Multivariate Cox proportional hazards analysis was employed to identify the most significant independent prognostic factors for PFS and OS. Actuarial survival rates were estimated using the Kaplan–Meier method, with statistical differences between groups assessed using the log-rank test. Additionally, the unpaired *t*-test was utilized to evaluate statistical significance of cell proliferation, while analysis of variance (ANOVA) followed by the Bonferroni correction for multiple comparisons was used to assess differences in cell migration between control and PR-A/PR-B groups in the ES-2 and TOV-21G cell lines. All experiments were conducted in triplicate, and data are presented as mean ± SD of triplicate experiments. Data management and analysis were performed using SPSS software version 22 (IBM, Armonk, NY, USA) and Prism software version 8.0 (GraphPad, Boston, MA, USA) for Windows, with statistical significance set at *p* < 0.05.

## 5. Conclusions

In conclusion, our findings suggest that tumors displaying weak PR expression are linked to heightened platinum resistance and potentially poorer survival outcomes, as revealed in both among people with OCCC and cell line models. Assessing PR status may enable clinicians to adopt a more precision-based approach for chemotherapy in people with OCCC. However, further investigations, including animal experiments and clinical studies, are warranted to elucidate the underlying mechanisms of PR in predicting chemosensitivity and its potential as a therapeutic target.

## Figures and Tables

**Figure 1 ijms-25-07942-f001:**
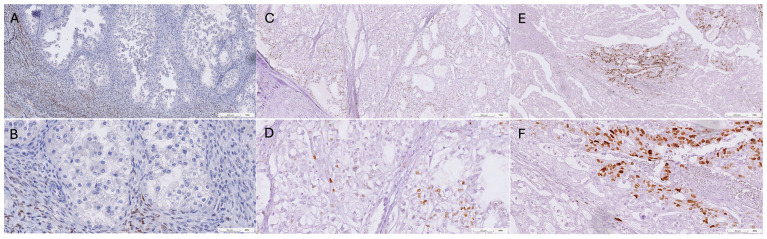
Illustrative examples of immunohistochemical staining for progesterone receptor protein in ovarian clear cell carcinoma tissues are shown as follows: (**A**,**B**) negative expression, (**C**,**D**) weak expression observed in 15% of tumor cells with an H score of 15, and (**E**,**F**) strong expression detected in 40% of tumor cells with an H score of 80. Upper rows (**A**,**C**,**E**), 100×, scale bar 200 µm. Lower rows (**B**,**D**,**F**), 400×, scale bar 50 µm.

**Figure 2 ijms-25-07942-f002:**
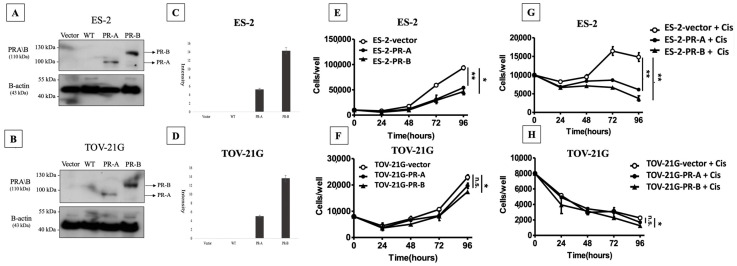
Western blot analysis revealed minimal expression of PR-A and PR-B proteins in wild-type ES-2 and TOV-21G cell lines (**A**,**B**). Following transfection with plasmids pcDNA3-hPR-A or pcDNA3-hPR-B, there was a substantial increase in the protein levels of PR-A or PR-B compared to their respective vector control cells (**C**,**D**). A comparison of cell proliferation across different PR isoform expressions (**E**,**F**). (**G**,**H**) Cisplatin (Cis) cytotoxicity experiment in ovarian clear cell carcinoma cells with varying PR isoform expressions. Control cells (represented by the circle line) were compared with PR-expressing cells (represented by the dot and triangle lines). Statistical significance is denoted by * for *p* < 0.05, ** for *p* < 0.01, and n.s. for nonsignificant differences.

**Figure 3 ijms-25-07942-f003:**
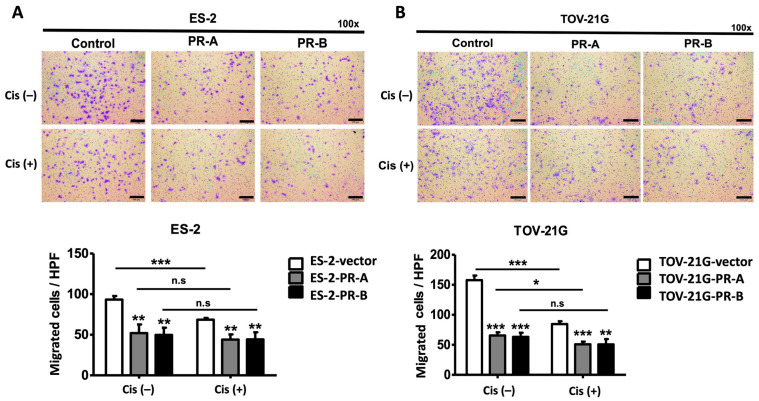
The migration assay conducted on ES-2 (**A**) and TOV-21G (**B**) cells utilizing the Transwell system revealed that PR-A- and PR-B-overexpressing cells exhibited reduced migration ability compared to vector cells. Upon treatment with cisplatin (Cis), the metastatic capacity was further inhibited in cells overexpressing PR-A or PR-B. Statistical significance is indicated by * for *p* < 0.05, ** for *p* < 0.01, *** for *p* < 0.001, and n.s. for nonsignificant differences. Scale bar 100 µm.

**Table 1 ijms-25-07942-t001:** Clinicopathological characteristics of all patients (N = 80).

Age, mean (SD)	48.9 (10)
Follow-up, months, median (range)	50 (5–132)
FIGO stage, n (%)	
I	44 (55)
II	12 (15)
III	20 (25)
IV	4 (5)
Menopause, n (%)	
Yes	41 (51.2)
No	39 (48.8)
Parity, n (%)	
Nulliparous	31 (38.8)
≥1	49 (61.2)
Pretreatment CA-125, n (%)	
<35 U/mL	11 (13.8)
≥35 U/mL	59 (73.8)
missing	10 (12.4)
Concurrent endometriosis, n (%)	
Yes	19 (23.8)
No	61 (76.2)
PR expression, n (%)	
Yes	30 (37.5)
No	50 (62.5)
PR H-score, mean (SD)	18.95 (34.73)
Platinum-sensitivity, n (%)	
Sensitive	66 (82.5)
Resistant	14 (17.5)

**Table 2 ijms-25-07942-t002:** Factors associated with platinum sensitivity in OCCC.

Factor	Platinum-Sensitive n = 66	Platinum-Resistant n = 14	*p* Value
Age (mean), years	48.3	51.7	0.441
Menopause, n (%)	32 (48.5)	9 (64.3)	0.283
Parity ≥ 1, n (%)	41 (62.1)	8 (57.1)	0.728
Concurrent endometriosis, n (%)	19 (28.8)	0 (0)	**0.033**
Pretreatment CA-125 (mean), U/mL	537	709	0.825
PR positive, n (%)	29 (43.9)	4 (28.6)	0.376
PR H score (mean)	21.9	5	**0.002**
PR H score ≥ 50, n (%)	13 (19.7)	0 (0)	0.110

*p* values were derived from two-sample *t*-test and chi-square or fisher’s exact test. Statistically significant *p* values are in bold.

**Table 3 ijms-25-07942-t003:** Factors associated with PR expression in OCCC.

Factor	PR^weak^ (n = 67)	PR^strong^ (n = 13)	*p* Value
Age (mean)	49.49	45.85	0.975
Menopause, n (%)	35 (52.2)	6 (46.2)	0.688
Parity ≥ 1, n (%)	42 (62.7)	7 (53.8)	0.549
FIGO stage			0.588
I (n = 44) (% within stage)	35 (79.5)	9 (20.5)	
II (n = 12) (% within stage)	10 (83.3)	2 (16.7)	
III (n = 20) (% within stage)	18 (90.0)	2 (10.0)	
IV (n = 4) (% within stage)	4 (100)	0 (0.0)	
Concurrent endometriosis, n (%)	17 (25.4)	2 (15.4)	0.439
Pretreatment CA-125 (mean)	573.7	532.2	0.324

PR^weak^, PR H score < 50; PR^strong^, PR H score ≥ 50. *p* values were derived from two-sample *t*-test and chi-square or fisher’s exact test.

**Table 4 ijms-25-07942-t004:** Multivariate Cox regression analyses of progression-free (PFS) and overall survival (OS).

	PFS			OS		
Factor	HR	95% CI	*p* Value	HR	95% CI	*p* Value
Menopause	1.324	0.515–3.403	0.560	1.940	0.707–5.318	0.198
Parity (≥1 vs. 0)	0.856	0.336–2.180	0.745	1.061	0.386–2.920	0.908
Concurrent endometriosis (yes vs. no)	0.141	0.018–1.084	0.060	0.187	0.024–1.464	0.110
CA-125 (≥35 vs. <35)	3.252	0.409–25.89	0.265	4.767	0.588–38.65	0.144
PR H-score (≥50 vs. <50)	0.369	0.085–1.594	0.182	0.311	0.070–1.391	0.126
FIGO stage (III/IV vs. I/II)	2.427	1.004–5.869	**0.049**	3.220	1.27–8.161	**0.014**

Statistically significant *p* values are in bold.

## Data Availability

All data of this study will be provided on reasonable request from the corresponding author.

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
