# Peer review of "How Progesterone Receptor Expression Impacts Platinum Sensitivity in Ovarian Clear Cell Carcinoma: Insights from Clinical and Experimental Perspectives"

_ijms, 2024, doi:10.3390/ijms25147942_

Round 1

Reviewer 1 Report (New Reviewer)

Comments and Suggestions for Authors

Type: Review

Title:

How Progesterone Receptor Expression Impacts Platinum Sensitivity in Ovarian Clear Cell Carcinoma: Insights from Clinical and Experimental Perspectives

In this review, the author put reasonable efforts into addressing the influence of progesterone receptor (PR) expression levels on platinum sensitivity and survival outcomes in OCCC patients using patient samples and cells lines. However, some issues preclude a recommendation for publication at this point, as described below.

1.    Figure 1: In panel B, the magnification is different. Please include a low power and a high power view for each panel.

2.    In the IHC run, did the author include a positive control? Please provide image.

3.    In line 83 “Strong PR expression was defined as a H-score of 50 or higher” while the Figure 1 legend, panel (C) said “Moderate expression detected in 40% of tumor cells with an H-score of 80”. Is H-score 80 moderate or strong expression?

4.    Line 87: “Univariate analysis revealed significantly improved 5-year PFS and OS for patients with early FIGO stage (I/II), concurrent endometriosis, and pretreatment CA125 levels lower than 35 U/mL, while strong PR expression exhibited only a trend towards improved outcomes.”. Can the authors provide a a table that shows these data?

5.    It is reported that ES-2 cells are resistant to cisplatin, whereas TOV21G cells are reported to be susceptible to it. Could the author explain why these two cell lines were selected and whether the difference in their cisplatin sensitivity status contributed to the results in Figure 2? What is the IC50 of cisplatin in these cells?

6.    Line 123: typo (Figure 1E & F): it should be (Figure 2E & F)

7.    In Materials and Methods, Cell viability section, the authors mentioned “cells were treated with cisplatin for 4, 24, 48, and 72 hours Figure 2 (G-H) did not show 4 hr data points. Further, it showed 96 hr data points. Please clarify.

Comments on the Quality of English Language

-

Author Response

Reviewer 2 Report (New Reviewer)

Comments and Suggestions for Authors

In this manuscript authors explored the influence of progesterone receptor (PR) expression levels on platinum sensitivity and survival outcomes in OCCC patients. Authors found that total PR H-score of platinum-sensitive tumors was significantly higher than that of platinum-resistant tumors. Moreover, PR protein was only weakly detectable in ES-2 and TOV-21G cells. Overexpression of PR-A or PR-B led to significantly reduced proliferation and migration in ES-2 and TOV-21G cells, and enhanced cisplatin cytotoxicity in these cell lines.

The manuscript is interesting and generally well written. Tables are easly readable but figures must be improved. See my comments below. 

Authors' affiliations must be completed 

Lines 50-52: It deserves to be pointed out that cisplatin resistance in ovarian cancer is mainly due to the increased antioxidant capacity of ovarian cancer cells (see PMID: 38203758 ). This is an important point to add since it can further highlight the interesting results obtained by the authors.

Figure 1:  "200X, scale bar 50um" must be removed form the image and insert in the figure legend

Western blot images: Insert kDa in the molecular weights

Lines 113-114: Author must load ES-2 and TOV-21G cell lysates in the same blot and report the densitometric analysis to state that "Immunoblot detection of PR-A and PR-B exhibited almost undetectable signals in 113 both ES-2 and TOV-21G wild-type cells" 

Figure 2C and D must be improved

Images in Figure 3A and B must be improved

Line 263: "1x" must be replaced with the correct molarity used

IHC quantification protocol and representative images for score 0 to 3 must be provided

4.6. Cell transwell migration assay: Which transwell authors used? product code must be provided

Abbreviations must be written in full length when mentioned for the first time

An accurate revision of typing errors is recommended

Author Response

Reviewer 3 Report (New Reviewer)

Comments and Suggestions for Authors

The strong point of this study is highlighting the influence of progesterone receptor expression on platinum sensitivity in ovarian clear cell carcinoma. However, some suggestions could improve the quality of the article:

-       What was the histological grading in the studied group?

-       Since you discussed PFS and OS in the multivariate analysis, what was the ratio of optimal versus suboptimal debulking at the level of the studied group?

-       In the Discussion section, additional information would be recommended regarding the fact that in postmenopause, when we observe an increased incidence of ovarian cancer, the protection due to progestin is absent.

 Kind regards

Round 2

Reviewer 1 Report (New Reviewer)

Comments and Suggestions for Authors

I thank the authors for addressing my comments.

Reviewer 2 Report (New Reviewer)

Comments and Suggestions for Authors

the manuscript has been significantly improved and can be accepted in the present form 

This manuscript is a resubmission of an earlier submission. The following is a list of the peer review reports and author responses from that submission.

Round 1

Reviewer 1 Report

Comments and Suggestions for Authors

This study aims to study the functional role of progesterone receptor genes A and B (PR1 and PRB) in the pathogenesis of OCCC. This study included clinical samples to support the conclusion of those OCCC patients with a lower expression of PR are associated with cisplatin-resistance by IHC. Then they use TOV21G cell model to demonstrate whether the expression of either PRA or PRB could sensitize to cisplatin-mediated cell cytotoxicity. The direction of this study is fine but the design and the result interpretation are poor. There are lots of pitfalls including the following main concerns;

1. The IHC analysis using H-score to determine the expression of PR with the clinicopathological parameters. It is fine to do statistical analysis. But there is no clear explanation of the calculation of the H-score? How many independent observers are to determine it? How to know which result of the A or B gene of PR? What's the controls of the antibody used? How to determine its specificity?

2. There's only one cell model, TOV21G, used in this study. This is the basic question of cell-to-cell biased problems.

3. This study lack enough functional and mechanistic study to conclude the reduced PR genes is associated with the cisplatin-resistance.

Author Response

We thank to the reviewer for the valuable comments, which allowed us an opportunity to improve our manuscript. All the issues raised by the reviewer were answered point-by-point in the attached file. We tried to reply every question rose by the reviewer as accurately as possible and also tried to make changes accordingly in our new version of the manuscript.

Reviewer 2 Report

Comments and Suggestions for Authors

Wu, CH. et al. aim to investigate the impact of progesterone receptor (PR) expression on platinum-sensitivity and survival outcomes in patients with OCCC. They collected and analyzed 592 patients with ovarian cancer. They also studied one OCCC cell line (TOV-21G). They concluded that patients with strong PR expression were associated with better platinum sensitivity and survival, and this was consistent with their experimental findings.

The manuscript is well written. Here are some concerns that hopefully will help to improve the overall manuscript.

(1)  Please provide a figure depicting the proposed model.

(2)  Do the authors perform experiments to analyze PR protein expression in samples from these patients?  Any evidence to prove PR expression in these human samples (weak or strong)?

Author Response

(The authors gave the same response as above.)

Round 2

Reviewer 1 Report

Comments and Suggestions for Authors

The revised version just clarified the questions raised previously. however, key suggestions, such as cell-to-cell biased data, and functional and mechanistic studies, were not provided. This could reduce the convincing value of the conclusions.